# Estimating the size distribution of plastics ingested by animals

Ifan B. Jâms[1], Fredric M. Windsor [1], Thomas Poudevigne-Durance[2], Steve J. Ormerod [1] & Isabelle Durance[1]✉

The ingestion of plastics appears to be widespread throughout the animal kingdom with risks to individuals, ecosystems and human health. Despite growing information on the location, abundance and size distribution of plastics in the environment, it cannot be assumed that any given animal will ingest all sizes of plastic encountered. Here, we use published data to develop an allometric relationship between plastic consumption and animal size to estimate the size distribution of plastics feasibly ingested by animals. Based on more than 2000 gut content analyses from animals ranging over three orders of magnitude in size (lengths 9 mm to 10 m), body length alone accounts for 42% of the variance in the length of plastic an animal may ingest and indicates a size ratio of roughly 20:1 between animal body length and the largest plastic the animal may ingest. We expect this work to improve global assessments of plastic pollution risk by introducing a quantifiable link between animals and the plastics they can ingest.

[1] Water Research Institute, Cardiff University, Wales CF10 3AX, UK. [2] School of Mathematics, Cardiff University, Wales CF24 4AG, UK.
✉email: durance@cardiff.ac.uk

I t is likely that plastics now flow through major food-webs across the Earth. Terrestrial, freshwater and marine food-webs are all at risk, with potential implications for individuals, populations and ecosystems, as well as human health[1–4]. More than 690 marine species[2,5] and over 50 freshwater species are reported to ingest plastic, with adverse effects postulated through physical damage, direct toxicity or physiological effects from additives and adsorbed pollutants[3,5]. Plastics may also alter the flux of energy and nutrients through both individual organisms and ecological networks[6]. The foundation for understanding all these processes is to quantify global primary plastic ingestion, and then the proliferation of plastics through food-webs. Although a key piece of the puzzle, current understanding of the entry and transfer pathways of plastics through food-webs is in its infancy[4].

The dynamics of plastics entering food-webs can be considered to adhere to two fundamental explanatory variables: (i) The extent to which plastics and animals physically co-occur in space and time; and (ii) the propensity of plastics to be ingested by an animal ('ingestibility'). To date, knowledge has largely been limited to the former, with calls for development of the latter[7]. Global distribution models of plastic pollution[8–11] and organisms[12–14] have enabled estimates of co-occurrence, including encounter rates, which are products of plastic and animal concentrations in space and time used to predict ecological risk[15]. Current understanding is limited by the unlikely assumption that all plastics are equally ingestible.

Variables likely to influence the ingestibility of plastic debris include feeding behaviour (e.g. those of filterers, visual predators, echolocators), the size distribution of prey items (for predatory animals), the colour of plastic particles[16], the degree of plastic degradation and the release of odorants and infochemicals (e.g. dimethyl sulphide)[17]. Understanding the processes affecting ingestibility require detailed life history and environmental information, making general predictions difficult and unpractical. Body size, however, is a simple metric that can be derived for any animal with minimal knowledge of its ecology and life history. Studies of allometry have repeatedly demonstrated the utility of body size for predicting complex biological characteristics[18,19].

Here, we collate a dataset on plastic ingestion by more than 2000 wild animals to generate an ecologically relevant, allometric relationship estimating the maximum size of plastic that any animal may ingest, based on an easily acquired metric: body length. In doing so, we generate information on the specific fraction of the global plastic pollution load that can be ingested by animals. This allows risk models of global plastic pollution to include biological information on the ingestibility of plastics, as well as established data on the physical co-occurrence of animals and plastics. Finally, in conjunction with co-occurrence data, we demonstrate the value of our approach for plastic pollution risk assessment in the natural environment.

## Results

**Descriptive power of the allometric relationship**. The animal-plastic size relationship (log10-log10 linear regression; $R^2 = 0.42$, $F_{1,63} = 46.06$, $p = 4.7e^{-09}$) presented in Fig. 1 relates the body length of an animal to the maximum length of plastic it can ingest; roughly by a ratio of 20:1. The underlying meta-analysis synthesises more than 2000 gut-content surveys of animals containing plastics. The animals ranged over three orders of magnitude in body length: from the common dragonet fish larvae (*Callionymus lyra*, body length: 9.00 mm) to the humpback whale (*Megaptera novaeangliae*, body length: 10.34 m). Data on ingested plastic for individual animals, from the same study, were grouped according to the lowest possible taxonomic rank (usually species: 91% of records, including one proposed species of fish yet to be

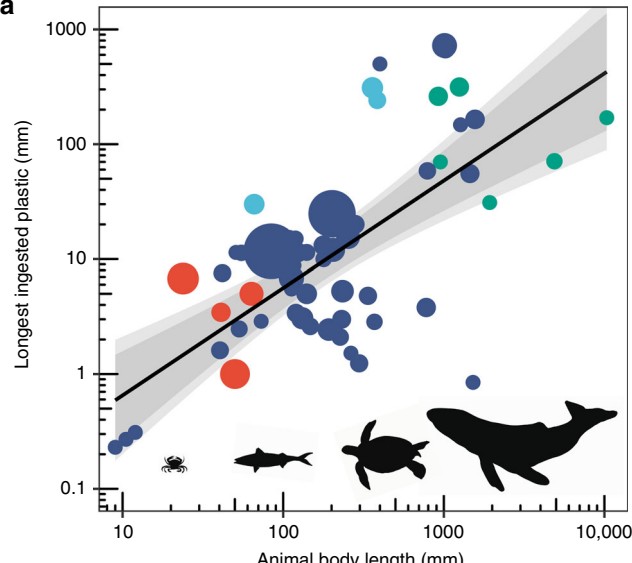

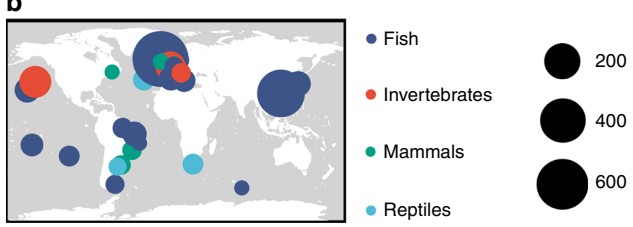

**Fig. 1 Allometric relationship between animal size and ingestible plastic size. a** Allometric size relationship (log10-log10 linear regression; $R^2 = 0.42$, $F_{1,63} = 46.06$, $p = 4.7e^{-09}$), including 99% (light grey) and 95% (dark grey) CIs, between animal body length (mm) and the longest piece of ingested plastic (mm) found during gut surveys (longest axis of largest piece of plastic found). Animal images are for illustration only and are not to scale. Each data point ($n = 65$) corresponds to the largest piece of plastic found within an animal taxon. **b** Distribution of field studies that provided data for the allometric relationship. Size of data points in **a** and **b** correspond to the number of individual animal specimens surveyed. Similar taxa from separate studies are plotted separately.

confirmed). This process made the most efficient use of the data available and provided data points likely to be closer to true values for an entire taxonomic population, rather than analyses of individual specimens.

$$\text{Plastic Size} = 10^{0.9341\log_{10}(\text{Body Size}) - 1.1200}$$

Records were predominantly for fish (Actinopterygii; 75%), followed by mammals (Mammalia; 9%), invertebrates (Polychaeta, Maxillopoda, Malacostraca, and Mollusca including the Bivalvia, Gastropoda and Cephalopoda; 11%) and reptiles (Reptilia; 5%). Species-level data (91% of records) were either for animals in marine environments only (42%), marine and brackish (25%), marine, brackish and freshwater (5%), freshwater and brackish (2%) or freshwater only (23%). No studies of terrestrial animals met the criteria for inclusion. The maximum reachable depths of the species-level records in this meta-analysis ranged from 25 m (Chinese mitten crab, *Eriocheir sinensis*) to 4000 m (humpback whale, *Megaptera novaeangliae*). These depth ranges far exceed the boundaries of current global models of plastic pollution distribution (Fig. 2).

All data were gathered via necropsy. A mixture of methods were observed, including the digestion of whole bodies and specific organs using chemical agents (including KOH, NaOH and $H_2O_2$)

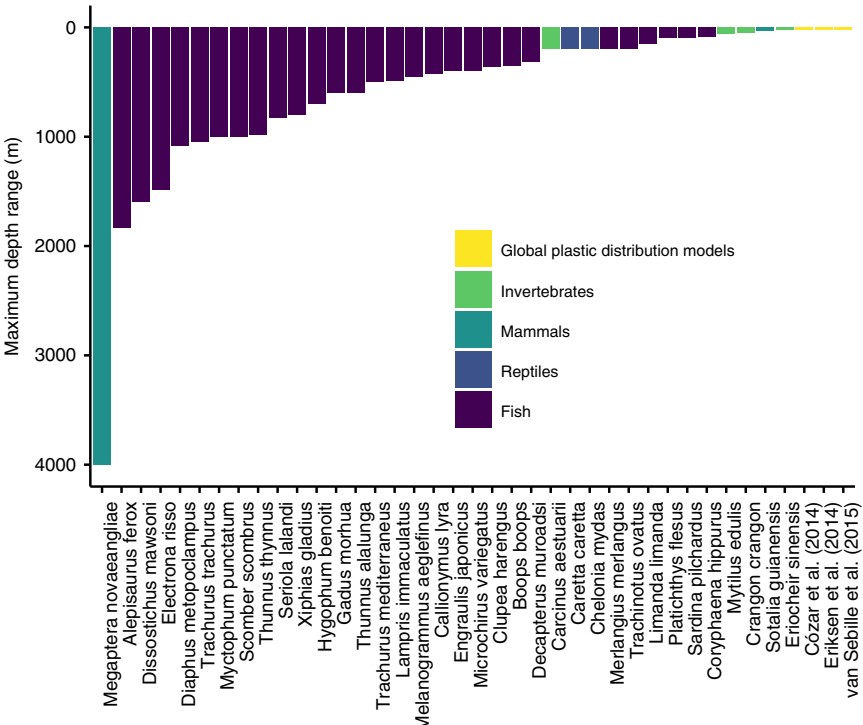

**Fig. 2 Animal depth ranges in the water column exceed plastic pollution models.** Water column depth range of species included in meta-analysis compared to depth range of established global plastic models. Cózar et al.[11], Eriksen et al.[8] and van Sebille et al.[10] are well cited distribution models of buoyant plastics floating at the surface of the Earth's oceans (674, 946 and 255 citations, respectively (Google Scholar 2019)). We assigned a crude water depth value of 25 m to each of these models, to account for the oceanic surface mixed layer, according to Kukulka et al.[31].

to identify plastics. Generally, studies on larger animals tended not to specify the use of a microscope during gut-content surveys (Fig. 3). The range of taxa involved, their biological traits, study locations and laboratory methods used combine to explain some of the deviation around the line of best fit.

**Predictive power of the allometric relationship.** Animal length alone explained 42% of the variance in the longest ingested plastic (animal-plastic size relationship, Fig. 1). We tested this relationship using a subset of observations for parameterisation (90%) and validation (10%), and we repeated the process 1000 times to compare the observed and validation data. Data deficiencies at both extremes of the relationship mean that predictions for both small (<30 mm) and large (>2000 mm) animals are less robust than for those between these extremes. This transpires as a general tendency to under-predict the size of plastic the largest animals may ingest, and to over-predict the size of plastics the smallest animals may ingest. This explains why the validation found roughly a third (30.45%) of observed values fell within 95% CIs. The predicted and observed data were similar (Root Mean Square of Errors (RMSE) = 0.68) and significantly related to one another, with reasonable explanatory power (Linear regression; $R^2 = 0.38$, $F_{1,5998} = 59.96$, $p < 0.001$).

**Challenges in identifying the smallest ingestible plastics.** Gut-content survey methods tend to scale with the size of the animal under study: studies of large animals in our dataset tended not to specify the use of a microscope during gut-content analyses (Fig. 3). While the largest piece of plastic ingested is often easily identified, locating the smallest fragment depends on the method used. Therefore, compared to large plastics, cross-study variations in gut-content methodologies reduce the reliability of data on the smallest ingestible plastics. While a positive correlation was

evident on log10-log10 scales, animal length accounted for little of the variance of the size of the smallest plastic fragment found in specimens (log10-log10 Linear Regression; $R^2 = 0.10$, $F_{1,61} = 7.58$, $p = 0.008$; Fig. 3). This is consistent with the hypothesis that plastics from a wide size spectrum up to the largest ingestible fragment are likely to be taken by animals.

**Application of the allometric relationship.** The taxonomic generality of the animal-plastic size relationship affords an array of applications. We illustrate just one, by mapping the risk of plastics entering the base of global food-webs: the zooplankton community (Fig. 4). We used the animal-plastic size relationship to select an appropriate size class of plastics the global zooplankton community may ingest; then created a risk map by combining ingestible plastic densities as provided by Eriksen et al.[8] with zooplankton densities provided by Strömberg et al.[20]. The increased accuracy of assessing the fraction of plastics zooplankton can ingest (Fig. 4a) can be seen clearly in comparison to the same risk map for all plastics in the oceans (Fig. 4b). Plastics entering the global zooplankton community have substantial potential for further trophic proliferation to a broad suite of species, including commercially important quarry. Areas of priority for mitigating the influx of plastics into global food-webs include the East and South China Seas, Bay of Bengal, Black, Mediterranean and Sargasso Seas, and European coasts of the north Atlantic Ocean.

**Discussion**

This study answers a recent call for strong and focused scientific research to guide international plastic pollution related initiatives[21]. Current global risk assessments of plastic pollution are limited to modelling the physical co-occurrence of plastics and animals in space and time. Here, we present a simple allometric

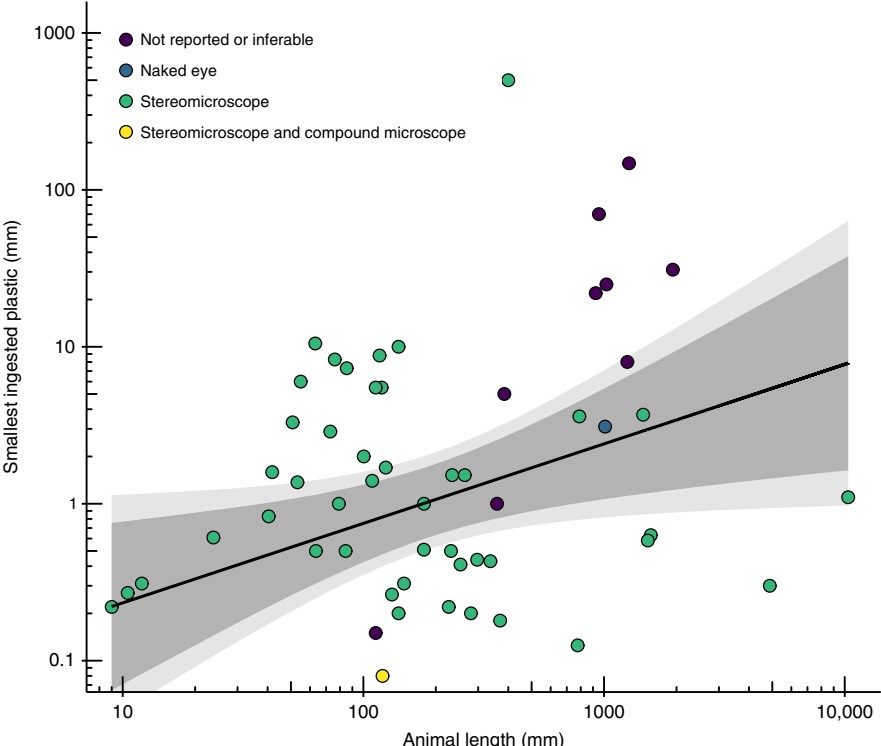

**Fig. 3 Detection limits scale with animal size.** Studies of larger animals tended not to specify the use of magnifying equipment. Weak relationship (log10-log10 linear regression; $R^2 = 0.10$, $F_{1,61} = 7.58$, $p = 0.008$), including 99% (light grey) and 95% (dark grey) CIs, between animals and the smallest piece of ingested plastic found during gut surveys (plastics measured along their longest axes; $n = 63$).

relationship to account for the ingestibility of plastics, and find that body length alone describes over 40% of the variance in the size of the largest plastic animals can ingest. This is an encouraging inroad for incorporating ecological variability (ingestibility) into coarse assessments of global plastic pollution risk.

Body size is important in many ecological processes[22,23], and has been demonstrated repeatedly as a useful predictor of secondary biological characteristics[18]. Previous studies have eluded to the influence of body size on the nature and frequency of plastic-biota interactions[4,24,25], but our study demonstrates the utility of this simple metric for predicting plastic consumption in ecological risk assessments. The obvious next steps will be to develop more sophisticated models targeting specific groups of animals by incorporating additional life history variables. Likely useful variables include feeding modes, mouthpart morphology, ontogeny and habitat preferences.

The simplicity of body size as a predictive metric in studies of plastic ingestion is attractive as it is applicable to any described (or undescribed) animal species. We emphasise, however, the absence of data from terrestrial animals, and their unique biological characteristics, in this meta-analysis. The current deficit of information on plastic pollution in terrestrial ecosystems, compared to marine environments, is a general problem in the field of plastic pollution research[4]. Addressing this deficit is a priority for developing accurate risk assessments and improving understanding of how, where, and through which pathways, plastics threaten global biota. We strongly encourage the publication of raw data and images alongside future studies of plastic pollution, which were often absent or incomplete.

Like the incorporation of ecological information into broad-scale assessments of plastic pollution risk, improved estimates of plastic distribution and concentration in the environment would also be valuable. Global models of plastic pollution are currently confined to the surface mixing zone of oceans[8,10]. Animals found

to contain plastics in this study are found at depths far greater (25–4000 m) than this surface mixing zone. Recent work in Monterey Bay, USA, found the greatest microplastic burden below the surface mixing zone[26]. More information is needed on the distribution and concentration of plastics at water depths where animals feed (but see Koelmans et al.[27]).

Overall, we find the size of a plastic particle relative to the size of an animal to be central to understanding the interactions between the two. Using the broadly applicable allometric relationship presented, this study draws attention to areas of concern regarding the flux of plastics into the base of the global aquatic food-web. The potential magnitude of plastic flux into this basal trophic layer is concerning for both environmental and human health. The generality of the plastic-animal relationship provides a foundation for developing a predictive understanding of the amount of plastic available for ingestion by any animal, and could be scaled to whole food-webs. Incorporating broadly applicable biological information into current global plastic pollution models presents a promising avenue for developing accurate risk assessments, and effective monitoring and mitigation efforts.

## Methods

**Systematic review.** On the 26th of January 2018 we used Web of Science (version 5.27) to find peer-reviewed research articles from 1900–2018 on plastic ingestion by any organism using the search string ((plastic OR plastics OR microplastic* OR mesoplastic* OR macroplastic*) AND (ingest* OR absorb* OR devour* OR eat* OR digest* OR consum* OR swallow* OR ingurgutat* OR engorg* OR gorge OR graz* OR masticat* OR ruminat* OR prey OR meal OR nourish* OR diet OR sustenance OR gastro* OR stomach* OR intest* OR assimili* OR incorporat* OR embod* OR engulf* OR envelop*) NOT (consumer)) under the heading "Topic". We searched Science Citation Index Expanded (SCI-EXPANDED) –1900-present; Social Sciences Citation Index (SSCI) –1956-present; Arts & Humanities Citation Index (A&HCI) –1975-present; Conference Proceedings Citation Index- Science (CPCI-S) –1990-present; Conference Proceedings Citation Index- Social Science & Humanities (CPCI-SSH) –1990-present; Emerging Sources Citation Index (ESCI) –2015-present.

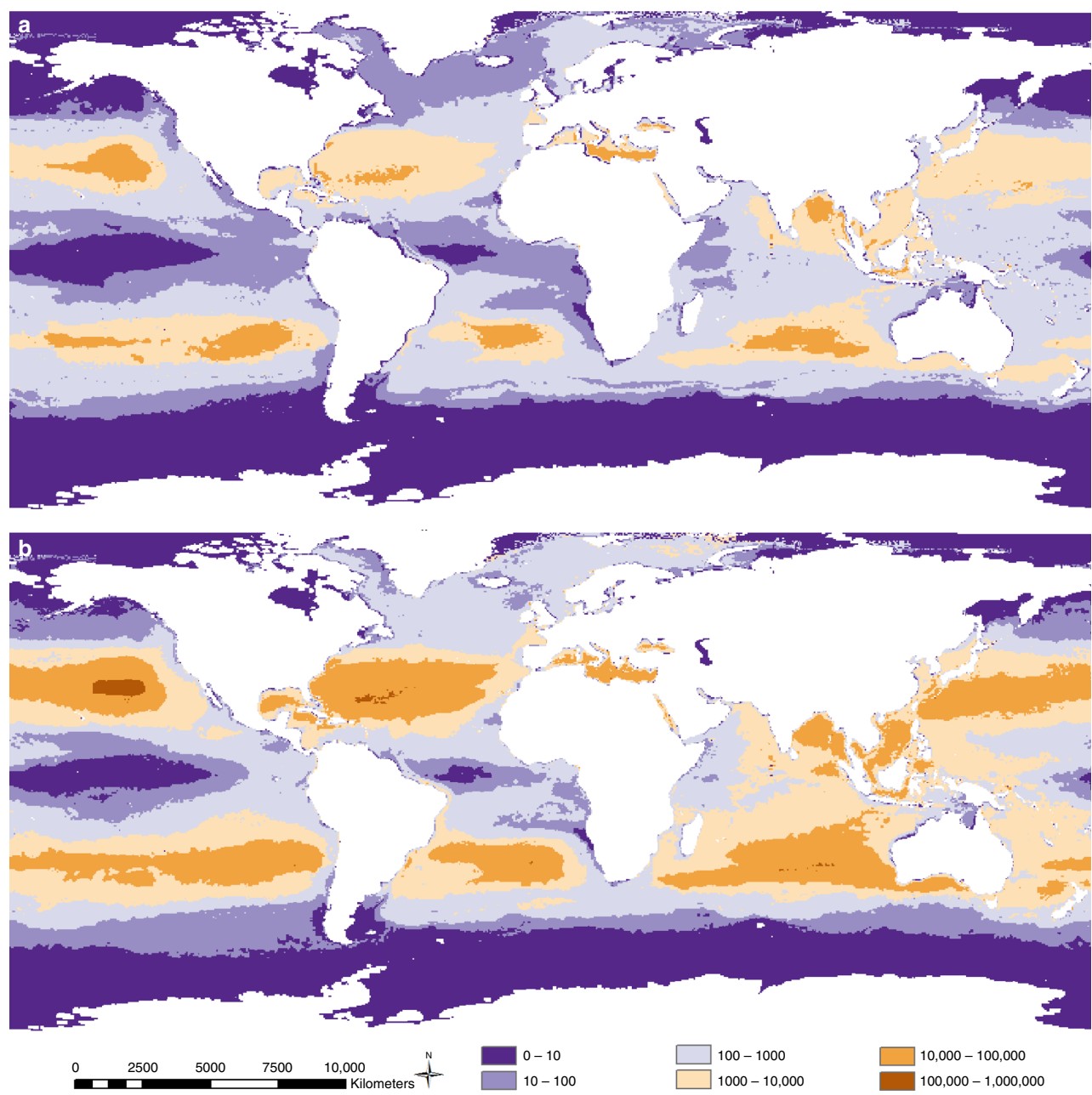

**Fig. 4 Global risk of plastic entering zooplankton communities. a** Using the animal-plastic size relationship to estimate the density of ingestible plastics (0.33–1.00 mm in length) divided by zooplankton density. **b** Current 'state of the art': as for **a**, with ingestible plastic densities substituted with total plastic densities. Legend coarsely estimates the level of plastic ingestion risk for zooplankton.

The 22,205 records found were listed using the "Relevance" function within Web of Science and the 1999 most relevant results were exported as a.txt file for screening. All 1999 titles were screened for relevance (title screening), and abstracts were also read in cases where the reviewer was unsure. In all cases where the reviewer was unsure, the article was retained for further screening at the next phase (full text screening). To be considered for the Data collection phase (below), an article was required to meet the following criteria: (1) Article seemed like it included some information on ingestion of any type or size of plastic by an organism; (2) article must report on field-based studies where plastics were present in the environment at natural concentrations and size distributions, as lab studies are often non-representative in terms of plastic availability. We excluded reports of plastic consumption by humans and reviews. Only peer-reviewed primary research articles were accepted. Articles found opportunistically after January 2018 were incorporated into the study according to the same inclusion criteria.

**Data collection**. Articles accepted for data collection reported or illustrated (e.g. via image analysis): (1) the size of the longest axis of ingested plastic (any plastic type) by a taxon of animal or a single animal; (2) the mean or mid-range body length of taxa or individuals containing ingested plastic. We also calculated the weighted-mean of mid-ranges or means provided for specific size-bins of body length as estimates of the wider pool of relevant taxa when the latter was unavailable. Data included as approximations of total body length were reported measurements of capitulum length, curved carapace length, and carapace width. The length of bivalves was recorded as the length of the shell. Descriptions of the exact dimensions measured of animal lengths were often unclear. Here we defined "total body length" as the distance from the most anterior to the most posterior part of an animal. Whether a measurement was classified as "total body length" usually required judgment of the methodology described.

Articles that provided plastic and animal size data for each specimen within a study were relatively sparse. More common were summary statistics for a group of individuals of the same species. Therefore, we prioritised the collection of data on animal species (i.e. groups of individuals). Where data were available for individual animals within a group as well as for the group as a whole, only data for the latter were retained to avoid pseudoreplication. Any data on individual animals were summarised for the lowest ranked taxonomic group possible. Data on single or smaller groups of individuals of greater taxonomic resolution were prioritised over summarised data for higher taxonomic levels.

Where data were available only for subgroups where different plastic measurements were made (e.g. in surface area in some individuals or in lengths for others), we used only data expressed as linear length. Where data were available only for a single animal, summary statistics for a group were replaced with the actual values recorded for that individual. The number of animals in a group was recorded. Where data on an animal were repeated in more than one study, we used the most precise data available only to avoid pseudoreplication.

The largest piece of plastic ingested by a group of individuals is likely to be more representative of the true maximum for an entire population than a single individual. Since matching a specific individual to a specific plastic fragment was seldom possible for groups of individuals, we used the mean body length in relation to a plastic fragment ingested by any group member. We focused on the precision of the relationship between body size and plastic size, by giving precedence to body length measurements of specimens that contained plastics (i.e. not all animals in a study would contain plastics), over data for wider groupings of animals (e.g. the mean body length of all animals in a study).

Ingested plastics were defined as those found in the main digestive tract of an animal via necropsy or tissue digestion. We excluded data on plastics in faeces or regurgitates, live animals or observations of plastic ingestion in behavioural studies. Regurgitated material might reflect material that could not be ingested further into the gastrointestinal tract while faeces would contain only those plastic fragments that could pass through the gastrointestinal tract and not be retained. Where available, the longest axis of the smallest ingested plastic fragment and the type of magnification used to detect plastics, was noted for each record. However, the absence of this information did not disqualify an article. The full set of collated data is provided with this article.

The use of reported values was prioritised, but in their absence, data were also collected on animal and plastic lengths from images using ImageJ (version 1.51J8). Measurements were made of the longest straight axis of a plastic fragment, calibrated according to the scale indicators in images, using a segmented line to measure long, coiled pieces of plastic material. Where coils of plastic could not be discriminated as a single piece, the maximum axis of the coil as a whole was measured. We used the image of highest resolution available. Only plastics that were wholly visible in an image were measured.

Decimal degree latitude and longitude coordinates were approximated from reported coordinates, or site descriptions where coordinates were unavailable, using Latlong. In the case of many sampling sites, an approximate central point was used for all sites in a study.

All species, including one proposed species of fish yet to be confirmed, were classified as "marine only", "marine and brackish", "marine, brackish and freshwater" or "freshwater only", according to FishBase[28] and SeaLifeBase[29]. Any records from SeaLifeBase were classified as "marine only" by default, with additional descriptions of tendencies for brackish or freshwater environments added to fit one of the four water type classifications. If available, data on depth range were also gathered from either FishBase or SeaLifeBase.

**Data analysis**. The universal allometric log10-log10 relationship between animal and ingestible plastic size was modelled and visualised as a linear regression using Microsoft Excel (version 16.16.7) and R (version 3.6.1; "Action of the toes"), within the RStudio environment (version 1.1.463).

**Validation of the allometric relationship**. To validate the plastic-animal size relationship (Fig. 1), we selected a subset of the data at random (10%) and a parameterisation dataset (90%). We repeated this procedure 1000 times to allow for a suitable understanding of the robustness of the allometric relationship and the potential limitations of this data for making predictions. In each instance, the parameterisation dataset was used to construct the allometric relationship, and predictions were made for the collated validation dataset. Predictions for the linear regression were constructed using the 'predict' function in the 'stats' package (version 3.4.3) in R (version 3.6.1; "Action of the toes"). We then used root mean square of errors (RMSE), in conjunction with a linear regression between predicted and observed values, to compare the predicted and observed data for the validation dataset to understand the relative accuracy of the plastic-animal size relationship for the 1000 simulated iterations.

**Plastic ingestion risk for zooplankton**. We used the global zooplankton distribution map provided by Strömberg et al.[20] to demonstrate the utility of the plastic-animal size relationship (Fig. 1). Strömberg et al.[20] combine primary production information with The Coastal and Oceanic Plankton Ecology, Production and Observation Database, COPEPOD[30], to produce a map of global zooplankton mass distribution. The body size range of the zooplankton represented is not provided in Strömberg et al.[20]. To gain this information, we downloaded biometric data for all organisms listed as "zooplankton" on COPEPOD and extracted the size range listed for all organisms. This dataset included measurements of total length and prosome length; the range found to be 0.12–13.5 mm. The animal-plastic size relationship provided herein estimates an animal 13.5 mm in length is able to ingest a piece of plastic 0.86 mm long.

We used the modelled global plastic distribution maps provided by Eriksen et al.[8] (Fig. 2 therein) that are separated into four plastic size classes: 0.33–1.00 mm,

1.00–4.76 mm, 4.76–20.00 mm and >20.00 mm. The four map images were imported into ArcGIS (version 10.5.1) in raster format, and georeferenced to a 10 m resolution ocean map sourced from Natural Earth Data (ne_10 m_ocean.shp, https://www.naturalearthdata.com/downloads/10m-physical-vectors/10m-ocean/). Interactive supervised classification of the images produced raster files, and the eight exponential classes of count concentration from 1 to 1,000,000, were transformed into a linear scale using $y = 10^{(6/8)x}$, which represented the maximum concentration of plastics found in that pixel.

We used the model prediction of global plastic count concentrations (number of pieces $km^{-2}$) for the size class (0.33–1.00 mm) as the fraction of plastic the zooplankton presented by Strömberg et al.[20] may ingest. Using this plastic size classification provides a buffer against underestimating environmental risk by: (1) Using the longest animal body length recorded to specify plastic length (as opposed to the mean size) and (2) exaggerating the maximum size of ingestible plastics to include those up to 1 mm, a buffer of 0.14 mm. To gain a map of all plastics present at the surface of the oceans and seas, we summed the values of the four size class maps presented by Eriksen and colleagues[8].

We processed the global zooplankton distribution map image provided by Strömberg et al.[20] in a similar way to the plastic maps. The image was imported into ArcGIS (version 10.5.1) in raster format, and georeferenced to the 10 m resolution ocean map sourced from Natural Earth Data. Interactive supervised classification of the image produced a raster file, and the five exponential classes of count density from 0 to 100 were transformed into a linear scale using $y = 10^{(2/5)x}$, to represent the maximum density of plankton found in that pixel. To understand the fraction of plastics zooplankton can ingest (Fig. 4a), we divided the number of plastic pieces (0.33–1.00 mm in length) by the density (mg C $m^{-3}$) of zooplankton, to map risk levels. A comparable risk map for all plastics in the oceans (Fig. 4b), was produced by dividing the total number of plastics (sum of all four size classes presented by Eriksen et al.[8]) by the density of zooplankton.

**Reporting summary**. Further information on experimental design is available in the Nature Research Reporting Summary linked to this paper.

## Data availability

All data collated and used in the study are available at https://github.com/fmwindsor/plastic-allometry.

## Code availability

All code used in the study are available at https://github.com/fmwindsor/plastic-allometry.

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

# ARTICLE

13. Birdlife International and Handbook of the Birds of the World. Bird species distribution maps of the world. Version 2018.1. (2018). http://datazone.birdlife.org/species/requestdis. (2019).
14. IUCN. The IUCN Red List of Threatened Species. Version 2016-1. (2016). http://www.iucnredlist.org. (2019).
15. Schuyler, Q. A. et al. Risk analysis reveals global hotspots for marine debris ingestion by sea turtles. Glob. Chang. Biol. **22**, 567–576 (2016).
16. Santos, R. G., Andrades, R., Fardim, L. M. & Martins, A. S. Marine debris ingestion and Thayer's law—the importance of plastic color. Environ. Pollut. **214**, 585–588 (2016).
17. Savoca, M. S., Wohlfeil, M. E., Ebeler, S. E. & Nevitt, G. A. Marine plastic debris emits a keystone infochemical for olfactory foraging seabirds. Sci. Adv. **2**, e1600395 (2016).
18. Peters, R. H. The ecological implications of body size. (Cambridge University Press, 1986).
19. Brown, J. H., Gillooly, J. F., Allen, A. P., Savage, V. M. & West, G. B. Toward a metabolic theory of ecology. Ecology **85**, 1771–1789 (2004).
20. Strömberg, K. H. P., Smyth, T. J., Allen, J. I., Pitois, S. & O'Brien, T. D. Estimation of global zooplankton biomass from satellite ocean colour. J. Mar. Syst. **78**, 18–27 (2009).
21. Haward, M. Plastic pollution of the world's seas and oceans as a contemporary challenge in ocean governance. Nat. Commun. **9**, 667 (2018).
22. Woodward, G. et al. Body size in ecological networks. Trends Ecol. Evol. **20**, 402–409 (2005).
23. West, G. B. & Brown, J. H. The origin of allometric scaling laws in biology from genomes to ecosystems: towards a quantitative unifying theory of biological structure and organization. J. Exp. Biol. **208**, 1575–1592 (2005).
24. Windsor, F. M., Tilley, R. M., Tyler, C. R. & Ormerod, S. J. Microplastic ingestion by riverine macroinvertebrates. Sci. Total Environ. **646**, 68–74 (2019).
25. Wilcox, C., Van Sebille, E. & Hardesty, B. D. Threat of plastic pollution to seabirds is global, pervasive, and increasing. Proc. Natl Acad. Sci. USA **112**, 11899–11904 (2015).
26. Choy, C. A. et al. The vertical distribution and biological transport of marine microplastics across the epipelagic and mesopelagic water column. Sci. Rep. **9**, 7843 (2019).
27. Koelmans, A. A., Kooi, M., Law, K. L. & van Sebille, E. All is not lost: deriving a top-down mass budget of plastic at sea. Environ. Res. Lett. **12**, 114028 (2017).
28. Froese, R., Pauly, D. & Eds. FishBase. www.fishbase.org. (2019).
29. Palomares, M. L. D., Pauly, D. & Eds. SeaLifeBase. www.sealifebase.org. (2019)
30. O'Brien, T. Copepod: A global plankton database. U.S. Dep. Commerce, NOAA Tech. Memo. www.st.nmfs.noaa.gov/copepod/. (2019).
31. Kukulka, T., Proskurowski, G., Morét-Ferguson, S., Meyer, D. W. & Law, K. L. The effect of wind mixing on the vertical distribution of buoyant plastic debris. Geophys. Res. Lett. **39**, L07601 (2012).

## Acknowledgements
We thank Mr Iwan Williams for sharing statistical expertise. F.M.W. was supported by a studentship from the GW4 + Doctoral Training Partnership Studentship funded by the Natural Environment Research Council (NE/L002434/).

## Author contributions
I.D. conceived the study with I.B.J. and F.M.W. I.B.J., F.M.W., T.P-D., S.J.O. and I.D. contributed to the design, implementation, analysis and writing of the manuscript. I.B.J. and F.M.W. contributed equally as lead authors.

## Competing interests
The authors declare no competing interests.
