## [Peer Review File · Nature Communications]

Reviewers' Comments:

Reviewer #1:

Remarks to the Author:

In this interesting short paper, the authors have constructed an allometric model to attempt to predict where the highest risks relating to ingestion of plastics might be for particular species, based on the ratio between the size of the animal and the size of the plastic. This is an interesting idea, and moves forwards from the assumption that just because plastic is at a particular location, it must be posing a risk.

The authors have used data from published studies to feed into their model and show that the greatest risks of ingestion occur at a ratio of 20:1 between an organism and the plastic it ingests.

The idea is interesting, but I wasn't clear from the text if the idea was that the risk to the organisms was greatest or the risk of ingestion was greatest. The text regarding turtle ingestion suggests it is the former, and that a larger sized plastic bottle was a greater risk to health than a smaller plastic particle. I don't see any evidence in the literature for that assumption, in fact there is discussion around an alternative view, that smaller particles are more likely to pass across biological barriers and cause internal effects.

I was also not clear what shape of particle was being considered. The text mentions only fragment and notes that the largest edge of the fragment is taken as the size metric. However, what happens for fibres? Over 50% of ingested plastic is actually in the form of fibres, and the length of fibre that can be found in some very small organisms is quite remarkable. I wonder if the same constraints that would apply to a spherical or fractal object would apply to long fibrous items/ synthetic ropes, etc.

It was also a bit confusing what was happening with the depth data. Was this a factor in the allometric equation, in which case the details are not clear enough to understand how. Or do they relate only to the distribution data for the map/figure?

In conclusion, an interesting concept but my feeling is that to be useful to others, there needs to be a more robust discussion and testing of the data taking into account the comments above

Reviewer #2:

Remarks to the Author:

The manuscript by Jâms et al. purports to develop a universal allometric relationship between animal length and the maximum size of microplastic (MP) particles that they can ingest. The authors surveyed 2,000 papers and extracted data on the size range of MPs ingested by 44 animal species of known size. They found that a log-linear model explained 46% of the variation between these animal lengths and the largest MP ingested, although there was a much weaker relationship between animal length and the smallest MP ingested (17% of variability explained). They compared the model predictions to a further 22 data points collected from studies published in 2019 and used coastal plastic concentrations and a range of plastic sizes captured by a neuston net to estimate the risk of ingestion by green turtles.

I enjoyed reading the manuscript and I believe the underlying idea is strongly merited and has great potential to improve our understanding of how organisms interact with MPs in nature. However, I have some major issues with the manuscript that prevent me from endorsing it.

My biggest concern centres on the quality of the underlying data. I appreciate that they were probably a huge amount of work to extract, but 44 data points are still a miniscule coverage of animal life in the marine and freshwater realms. The range of body masses that they span is impressive, but I would worry about drawing "universal" conclusions from such a small percentage of total diversity. It is telling that the authors found 50% as much data in 2019 alone for their "model validation" exercise (Table 1), suggesting that there will be a lot more data coming online in the coming years. I fear that this study comes a little too soon to be an adequate meta-analysis on the topic.

Similarly, just because the authors noted the largest (and smallest) MP consumed by a particular species in one study, this does not in any way indicate that it is the maximum (or minimum) MP that the organism can consume. If each data point was informed by say 10 or more studies, then I would have a lot more confidence in the range of MPs that are attributed to each species.

The confirmation that these concerns are valid comes in Figure 1, where there is a positive correlation between animal length and maximum MP size, but the variation is huge (keeping in mind that this is a log₁₀-log₁₀ scale). I count 22 data points (out of 44) that fall outside the 99% CI of the model. Basically, this model should do a really poor job of predicting the maximum MP ingested by an animal of given size around 50% of the time.

The reality is even worse. When the authors compare their model to the 22 additional data points that they collected from studies published in 2019, they find that only 32% of the maximum MPs ingested by those new species fall within the 99% CI of their model predictions and more than half of those don't even fit within the 95% CI of their model. Worse still, all but two data points out of the 68% of new data that don't fit with their model predictions are larger than the maximum body size predicted by the model. So if I were a conservation manager and wanted to use this model to predict the maximum MP size that my target species for conservation could consume (e.g. to inform the regulations I impose on a protected area), there is an extremely high probability that I would underestimate the size of the plastic particles that could be ingested by the organism.

My final gripe with the study is that the authors use a case study of green sea turtles to illustrate the utility of their "universal" equation, but don't actually apply the equation in determining the potential risk for turtles. I would have expected them to use the 95% CI or 99% CI from the models to estimate the range of MP sizes that could be consumed by green turtles and then use this to inform the cut-off for plastic concentrations in the sea water occupied by the turtles. Instead, they use an entirely subjective size range informed by a neuston net, which has nothing to do with their model. In other words, Figure 3 could just as easily have been produced without ever needing to determine a relationship between animal size and minimum/maximum MPs ingested.

I would recommend that the authors at the very least combine their two datasets (i.e. 66 data points) to describe more of the variability in the true relationship between animal size and MPs ingested. I suspect they will need to do a lot more work to build on this database over time, however. A consideration that could help them push this into a higher tier would be to start generating their own data by testing the range of MP particle sizes that various animals could ingest. This would remove the troublesome doubt about whether these studies actually describe the true range of particle sizes that each animal species can ingest.

Some minor comments:

Ln65. Change to "its body length"

Ln89. The reference to Figure 2b does not seem appropriate here. This is a figure of animal length to smallest plastic particle ingested, not a figure summarising the use of microscopes during gut-content

surveys as the text suggests.

Ln102. This sentence entirely repeats the one at Ln89 and again, the citation to Figure 2b is completely inappropriate.

Ln108. If there is a significant relationship, then you should visualise the line of best fit (with confidence intervals) in Figure 2b, even if the model explains a small amount of variability.

Ln116. I assume the 95% CI here are meant to be 19–82, rather than the odd [19, 82] term that looks more like a citation? Why not use these 95% CI (or even 99% CI) to determine the risk of green turtles ingesting plastic, rather than the completely arbitrary dimensions of a neuston net (Ln283)? Or more correctly, you should be using the upper 95% CI of the maximum MP size model and the lower 95% CI of the minimum MP size model to determine the plausible range of MP particles that could be consumed by the turtles.

Ln166. Change to “short-term”

Table 1. Choose whether to present the 95% CI or 99% CI throughout and stick with that. It makes no sense to present both. In fact, it damages your case here because we can clearly see that most of these data either fall outside the 99% CI or barely scrape into them (and not the 95% CI).

**Response to reviewers' comments**

**General Response**

We are very grateful for the time both Reviewer 1 and 2 invested in this manuscript. We feel
Reviewer 1's issues with the study stem from an initial misinterpretation of what was done. This
has been valuable feedback - the previous draft plainly was not clear enough. We have simplified
the text throughout to ensure the piece's suitability for a wide audience. Reviewer 2's engagement
with the work was highly encouraging, and all of the main suggestions made by this reviewer have
been implemented and incorporated, markedly improving the quality of the manuscript.

**Specific Responses**

Responses to specific comments appear in red.

*Reviewer #1 (Remarks to the Author):*

In this interesting short paper, the authors have constructed an allometric model to attempt to
predict where the highest risks relating to ingestion of plastics might be for particular species,
based on the ratio between the size of the animal and the size of the plastic. this is an interesting
idea, and moves forwards from the assumption that just because plastic is at a particular location,
it must be posing a risk.

The authors have used data from published studies to feed into their model and show that the
greatest risks of ingestion occur at a ratio of 20:1 between an organism and the plastic it ingests.

This, we think, is where the central accomplishment of the study was not explicit enough, as is
evident from Reviewer 1's comment. To clarify: the animal-plastic size relationship does not:

*“show the greatest risks of ingestion occur at a ratio of 20:1 between an organism and the plastic*
*it ingests.”* We have simplified the text throughout and are now confident it will be obvious to the
reader that the 20:1 ratio estimates the maximum length of plastic any animal may ingest, based
on the animal's length.

Specific examples now in the text include:

Lines 54-56: *“We address this problem by presenting a single allometric relationship for estimating*
*the maximum size of plastic any animal may ingest.”*

Lines-89-91: *“Here, we collate a dataset on plastic ingestion by more than 2000 wild animals to*
*generate an ecologically relevant, allometric relationship estimating the maximum size of plastic*
*that any animal may ingest, based on an easily acquired metric: body length.”*

Lines 164-166: *“Body length alone describes over 40% of the variance in the size of the largest*
*plastic animals can ingest.”*

The idea is interesting, but I wasn't clear from the text if the idea was that the risk to the organisms
was greatest or the risk of ingestion was greatest. The text regarding turtle ingestion suggests it
is the former, and that a larger sized plastic bottle was a greater risk to health than a smaller
plastic particle. I don't see any evidence in the literature for that assumption, in fact there is
discussion around an alternative view, that smaller particles are more likely to pass across
biological barriers and cause internal effects.

The issue raised here is related to the initial misunderstanding by Reviewer 1, which has been
addressed above. The confusion Reviewer 1 found over the turtle study-case should now be
remedied by a wholesale change. The manuscript presents an entirely new demonstration of the
utility of the work in relation to the global zooplankton community. We are proud of this substantial
addition, which bridges the gap between plastic pollution modellers and ecologists.

Box 1 | Application of the allometric relationship

The taxonomic generality of the animal-plastic size relationship affords an array of applications. We illustrate just one, by mapping the risk of plastics entering the base of global food webs: the zooplankton community (Fig. 4). We used the animal-plastic size relationship to select an appropriate size class of plastics the global zooplankton community may ingest; then created a risk map by combining *ingestible* plastic densities as provided by Eriksen and colleagues⁸ with zooplankton densities provided by Strömberg *et al.*²⁰. The increased accuracy of assessing the fraction of plastics zooplankton *can ingest* (Fig. 4a) can be seen clearly in comparison to the same risk map for *all* plastics in the oceans (Fig. 4b). Plastics entering the global zooplankton community have substantial potential for further trophic proliferation to a broad suite of species, including commercially important quarry. Areas of priority for mitigating the influx of plastics into global food webs include the East and South China Seas, Bay of Bengal, Black, Mediterranean and Sargasso Seas, and European coasts of the north Atlantic Ocean.

Figure 4 | Global risk of plastic entering basal zooplankton communities. (a) Using the animal-plastic size relationship to estimate the density of *ingestible plastics* (0.33-1.00 mm in length) divided by zooplankton density. **(b)** Current approach: as for **a**, with *ingestible plastic*

densities substituted with *total plastic* densities. Legend coarsely estimates the level of plastic ingestion risk for zooplankton.

I was also not clear what shape of particle was being considered. The text mentions only fragment
and notes that the largest edge of the fragment is taken as the size metric. However, what
happens for fibres? Over 50% of ingested plastic is actually in the form of fibres, and the length
of fibre that can be found in some very small organisms is quite remarkable. I wonder if the same
constraints that would apply to a spherical or fractal object would apply to long fibrous items/
synthetic ropes, etc.

This is a valid point, and one we considered in depth when embarking on this work. We emphasise
throughout the amended draft, that data were collected on the longest axis, of the largest piece
of plastic found within an animal. This, of course, does mean that the length of ingested fibres is
captured in the dataset. One of the strengths of the work stems from the great range of animal
sizes studied. The magnitude of this size range, considered on a log₁₀-log₁₀ scale, means that
the relatively small differences in the length of fibres compared to fragments are minimised. The
resultant animal-plastic length relationship provides a general foundation for predicting plastic
ingestion, upon which specific anomalies, such as the ones Reviewer 1 refers to, can be built and
investigated.

These considerations are presented in detail in the Methods section, including:

Lines 234-235: “Articles accepted for data collection reported or illustrated (e.g. via image
analysis): (1) the size of the longest axis of ingested plastic (any plastic type) by a taxon of animal
or a single animal...”

Lines 267-271: “Ingested plastics were defined as those found in the main digestive tract of an
animal via necropsy or tissue digestion. We excluded data on plastics in faeces or regurgitates,

live animals or observations of plastic ingestion in behavioural studies. Regurgitated material
might reflect material that could not be ingested further into the gastrointestinal tract while faeces
would contain only those plastic fragments that could pass through the gastrointestinal tract and
not be retained.”

Lines 276-281: “The use of reported values was prioritised, but in their absence, data were also
collected on animal and plastic lengths from images using ImageJ (version 1.51J8).
Measurements were made of the longest straight axis of a plastic fragment, calibrated according
to the scale indicators in images, using a segmented line to measure long, coiled pieces of plastic
material. Where coils of plastic could not be discriminated as a single piece, the maximum axis of
the coil as a whole was measured.”

It was also a bit confusing what was happening with the depth data. was this a factor in the
allometric equation, in which case the details are not clear enough to understand how. or do they
relate only to the distribution data for the map/figure?

To clarify – depth data were not included in the animal-plastic relationship. The plotted depth data
illustrate the limitation of currently available plastic pollution models only. This fact has been
clarified throughout, and the depth plot restyled as a single figure to avoid confusion. The code
for modelling the simple animal-plastic size relationship using the software *R* is also provided,
which allows readers to check the analysis, if they so wish.

Examples of this in the text include:

Lines 115-118: “The maximum reachable depths of the species-level records in this meta-analysis
ranged from 25 m (Chinese mitten crab, *Eriocheir sinensis*) to 4000 m (humpback whale,
*Megaptera novaeangliae*). These depth ranges far exceed the boundaries of current global
models of plastic pollution distribution (Fig. 2).”

Lines 293-295: “The universal allometric log₁₀-log₁₀ relationship between animal and ingestible
plastic size was modelled and visualised as a linear regression using Microsoft Excel version
16.16.7 and R (version 3.6.1; “Action of the toes”), within the RStudio environment (version
1.1.463).”

In conclusion, an interesting concept but my feeling is that to be useful to others, there needs to
be a more robust discussion and testing of the data taking into account the comments above

We thank Reviewer 1 for the encouraging sentiment; and regarding the request to “*discuss*” and
“*take into account comments above*”, we hope our explanation of the source, and amendments
to resolve, the miscommunication pointed out by Reviewer 1 to be sufficient. Regarding the
request for greater testing, we refer to all-new analyses described in the “Predictive power of the
allometric relationship” and “Validation of the allometric relationship” passages of the Main Text
and Methods sections, respectively.

New passages:

Lines 135-147: “**Predictive power of the allometric relationship**

Animal length alone explained 41% of the variance in the longest ingested plastic (animal-plastic
size relationship, Fig. 1). We tested this relationship using a subset of observations for
parameterisation (90%) and validation (10%), and repeated the process 1000 times to compare
the observed data with the validation data. All validation data fell within the upper and lower 95%
confidence intervals. It is noteworthy that, for plastics longer than 100 mm, the animal-plastic size
relationship under-predicts the length of the longest ingested plastic, making estimates
conservative. Data deficiencies at both extremes of the equation means that predictions for both
small (<1 mm) and large (>150 mm) animals are less robust than for those between these
extremes. The predicted and observed data were similar (Root Mean Square of Errors
(RMSE) = 0.68) and significantly related to one another, with reasonable explanatory power

($R^2 = 0.38$, $F_{1,5998} = 59.96$, $p < 0.001$), suggesting predictions based on the animal-plastic size
relationship are reliable.”

Lines 296-306: **“Validation of the allometric relationship**

To validate the plastic-animal size relationship (Fig. 1), we selected a subset of the data at random
(10%) and a parameterisation dataset (90%). We repeated this procedure 1000 times to allow for
a suitable understanding of the robustness of the allometric relationship and the potential
limitations of this data for making predictions. In each instance, the parameterisation dataset was
used to construct the allometric relationship, and predictions were made for the collated validation
dataset. Predictions for the linear regression were constructed using the ‘predict’ function in the
‘stats’ package (version 3.4.3) in R (version 3.6.3; ‘Kite-Eating Tree’). We then used root mean
square of errors (RMSE), in conjunction with a linear regression between predicted and observed
values, to compare the predicted and observed data for the validation dataset to understand the
relative accuracy of the plastic-animal size relationship for the 1000 simulated iterations.”

*Reviewer #2 (Remarks to the Author):*

The manuscript by Jâms et al. purports to develop a universal allometric relationship between
animal length and the maximum size of microplastic (MP) particles that they can ingest. The
authors surveyed 2,000 papers and extracted data on the size range of MPs ingested by 44
animal species of known size. They found that a log-linear model explained 46% of the variation
between these animal lengths and the largest MP ingested, although there was a much weaker
relationship between animal length and the smallest MP ingested (17% of variability explained).
They compared the model predictions to a further 22 data points collected from studies published
in 2019 and used coastal plastic concentrations and a range of plastic sizes captured by a neuston
net to estimate the risk of ingestion by green turtles.

We are buoyed by Reviewer 2's interpretation of the study; and take the liberty of assuming that
Reviewer 2 is simply referring to plastics as microplastics (MPs) out of habit here, and has not
truly misunderstood the scope of this study. Microplastics are generally accepted as those under
5 mm in length. This study considers all plastics, including micro-, meso- and macroplastics.

See, for example:

Lines 89-91: "Here, we collate a dataset on plastic ingestion by more than 2000 wild animals to
generate an ecologically relevant, allometric relationship estimating the maximum size of plastic
that any animal may ingest, based on an easily acquired metric: body length."

Line 109: Figure 1 (Note Y axis "Longest ingested plastic (mm)" displays plastics on a scale that
ranges 0.1-1000 mm. Plotted data include plastics shorter than 5 mm (microplastics), but only as
a portion of the wider size distribution of plastic fragments present in the environment.

$$\text{Plastic Size} = 10^{0.9341 \log_{10}(\text{Body Size}) - 1.1200}$$

**“Figure 1 | Allometric relationship between animal size and ingestible plastic size. (a)** Allometric size relationship
 **(log₁₀-log₁₀; R² = 0.42, F_{1,63} = 46.06, p < 0.001), including 95% and 99% CIs, between animal body length (mm) and**
 **the longest piece of ingested plastic (mm) found during gut surveys (longest axis of largest piece of plastic found).**
 **Animal images for illustration only and not owned by the authors. Each data point corresponds to the largest piece of**
 **plastic found within an animal taxon. (b) Distribution of field studies that provided data for the allometric relationship.**

Size of data points in **a** and **b** correspond to the number of individual animal specimens surveyed. Similar taxa from
separate studies are plotted separately."

I enjoyed reading the manuscript and I believe the underlying idea is strongly merited and has
great potential to improve our understanding of how organisms interact with MPs in nature.
However, I have some major issues with the manuscript that prevent me from endorsing it.

We are pleased this piece was an enjoyable read, and are reassured to find Reviewer 2's
assessment of the work's merit to match ours – both for the underlying scientific rigour and as
evidence of the narrative's capacity to communicate the findings of this study to the reader. We
reiterate our assumption regarding the term MPs, and will continue with this assumption for the
remainder of this letter. We found all of the major issues raised by Reviewer 2 to be 'fixable', and
detail the implementation of those improvements in the following responses.

My biggest concern centres on the quality of the underlying data. I appreciate that they were
probably a huge amount of work to extract, but 44 data points are still a miniscule coverage of
animal life in the marine and freshwater realms. The range of body masses that they span is
impressive, but I would worry about drawing "universal" conclusions from such a small percentage
of total diversity. It is telling that the authors found 50% as much data in 2019 alone for their
"model validation" exercise (Table 1), suggesting that there will be a lot more data coming online
in the coming years. I fear that this study comes a little too soon to be an adequate meta-analysis
on the topic.

Reviewer 2 raises a good point – and we agree. The term "universal" has been removed from the
text – and we are thankful to the Reviewer for highlighting the issue.

Reviewer 2's issue with the timeliness of this study, we think, stems from a miscommunication on
our part as to the nature of the data. We read the above comment as taking issue with data

*quantity* and representativity; and not, strictly with *quality*. As such, we regard this concern related
to the comment that follows, and address both below.

Similarly, just because the authors noted the largest (and smallest) MP consumed by a particular
species in one study, this does not in any way indicate that it is the maximum (or minimum) MP
that the organism can consume. If each data point was informed by say 10 or more studies, then
I would have a lot more confidence in the range of MPs that are attributed to each species.

This we found a very valuable insight. Each data point *is*, indeed, informed by often hundreds of
individual animals. We share Reviewer 2's bolstered confidence in these data wholeheartedly,
when they are considered as what they are – summaries of up to 761 individual animals
represented by each data point. This is an important ecological, as well as statistical point raised
by Reviewer 2, and was subject to substantial consideration during study design. When designing
the study, we concluded that the largest piece of plastic ingested by a group of individuals is likely
to be more representative of the true maximum for an entire population than a single individual;
and also made best use of the data available in the literature.

In addition, the animal-plastic size relationship is now based on an extended dataset, by utilising
the approach Reviewer 2 suggested in later comments. We are grateful for this feedback, which
motivated more efficient use of the data. It is reassuring that the animal-plastic size relationship
maintained its general characteristics when updated to include more data.

We apologise if we misinterpret Reviewer 2's concerns. If they are, in fact, related to the perennial
debate around how much data is required to tackle urgent environmental challenges, we can only
state the obvious: we are confident our work strikes a good balance given the urgency of the
plastic pollution challenge. Certainly, this work is a major advancement from the current state of
affairs, whereby ecology informs little on global plastic pollution models. As with, for example, the
early work on global climate change, we believe it is important to publish works on plastic pollution

for scrutiny by the scientific community. The readers of *Nature Communications* are, of course,
well placed to contextualise these findings, and we look forward to progressing the discourse
around this pervasive issue.

Again, we share Reviewer 2's confidence in the merit of this meta-analysis. The probability of
"*more data coming online in the coming years*" is precisely why it's imperative this study is
published now: one of the key findings is the need for standardised reporting of plastic ingestion
data, to facilitate synthetic analyses as more data become available. The high visibility and
authority of *Nature Communications* makes it the ideal journal to draw attention to this need, and
we anticipate this facet of the manuscript will generate high citation rates.

Updates to illustrate the nature of the data more effectively include:

Line 109: Figure 1 now represents the number of individual animals that represent each data point
by the size of the data points.

$$\text{Plastic Size} = 10^{0.9341 \log_{10}(\text{Body Size}) - 1.1200}$$

**“Figure 1 | Allometric relationship between animal size and ingestible plastic size. (a)** Allometric size relationship
 **(log10-log10; $R^2 = 0.42$, $F_{1,63} = 46.06$, $p < 0.001$), including 95% and 99% CIs, between **animal body length (mm)** and**
 **the **longest piece of ingested plastic (mm)** found during gut surveys (longest axis of largest piece of plastic found).**
 **Animal images for illustration only and not owned by the authors. Each data point corresponds to the largest piece of**
 **plastic found within an animal taxon. (b)** Distribution of field studies that provided data for the allometric relationship.

Size of data points in **a** and **b** correspond to the number of individual animal specimens surveyed. Similar taxa from
separate studies are plotted separately.”

Lines 259-266: “The largest piece of plastic ingested by a group of individuals is likely to be more
representative of the true maximum for an entire population than a single individual. Since
matching a specific individual to a specific plastic fragment was seldom possible for groups of
individuals, we used the mean body length in relation to a plastic fragment ingested by any group
member. We focused on the precision of the relationship between body size and plastic size, by
giving precedence to body length measurements of specimens that contained plastics (i.e. not all
animals in a study would contain plastics), over data for wider groupings of animals (e.g. the mean
body length of all animals in a study).”

Use of extended and updated data detailed on:

Lines 296-306 “**Validation of the allometric relationship**” (passage included above).

The confirmation that these concerns are valid comes in Figure 1, where there is a positive
correlation between animal length and maximum MP size, but the variation is huge (keeping in
mind that this is a log₁₀-log₁₀ scale). I count 22 data points (out of 44) that fall outside the 99%
CI of the model. Basically, this model should do a really poor job of predicting the maximum MP
ingested by an animal of given size around 50% of the time.

We agree that there are plenty of opportunities for developing more complex models, which
explain a greater proportion of variance around the trendline. On the other hand, it is remarkable
to consider the breadth of life history and geography accounted for by one, simple, easily attained
variable (animal body length), and that this one variable explains close to half of the variance
observed.

In simple terms, this is a ‘*glass half full or half empty*’ discussion, for which the appropriate forum
is elsewhere. But for context, we briefly clarify our position. We maintain: a model composed of a

single explanatory variable, easily attainable for any animal, that explains close to half of the
variance observed; is an enviable foundation for developing more targeted models. Serious
assessments of plastic as a global pollutant requires this kind of functional foundation, to cut
through taxonomic constraints. Against the background of high rates of environmental change,
there is a widely-held realisation, with which we venture Reviewer 2 would agree: ecology and
the multidisciplinary field of ecosystem science must progress from a descriptive discipline into a
predictive one. We no longer have the time to measure and describe each leaf, thorn and grain
of sand on the *Entangled Bank* (Currie 2019). Contemporary rates of environmental change
require a Newtonian understanding of underlying mechanics. This study takes the latter approach.
We ask: what factor(s) can statistically account for the observed variance of nature? The finding
is empowering.

Regardless; this study certainly is a major advancement from the current state of affairs, which is
based on crude physical co-occurrence models devoid of biological mechanisms. We share
Reviewer 2's aspiration for greater predictive power. However, the fact that the animal-plastic
relationship is not a perfect analogue of nature does not, of course, discount its usefulness.
Publishing the animal-plastic relationship and its characteristics is important to accelerate its
improvement and flexibility. Detailed life-history studies of single taxa are sure to yield greater
precision than general, cross-taxa trends. However, this does not, for example, inform on the vast
array of undescribed species. The animal-plastic size relationship allows quantitative
approximations to be made with the scantest ecological data, which is an important advancement
from qualitative estimations. We present this work as a general foundation, which serves even
those undescribed taxa, and is suitable for development into more sophisticated models that
account for life histories, localities and so on.

Currie, D.J., 2019. Where Newton might have taken ecology. *Global ecology and biogeography*,
28(1), pp.18-27.

The reality is even worse. When the authors compare their model to the 22 additional data points
that they collected from studies published in 2019, they find that only 32% of the maximum MPs
ingested by those new species fall within the 99% CI of their model predictions and more than
half of those don't even fit within the 95% CI of their model. Worse still, all but two data points out
of the 68% of new data that don't fit with their model predictions are larger than the maximum
body size predicted by the model. So if I were a conservation manager and wanted to use this
model to predict the maximum MP size that my target species for conservation could consume
(e.g. to inform the regulations I impose on a protected area), there is an extremely high probability
that I would underestimate the size of the plastic particles that could be ingested by the organism.

*Reviewer 2 makes a valid point regarding conservative predictions. We detail the characteristics*
*and limitations of these predictions in the text, explaining that data-deficiencies at the extremes*
*of the animal size spectrum coincide with more conservative predictions. Beyond this, we are*
*returned to the *glass half-full or half-empty* discussion. We trust our position on this is clear from*
*the response above.*

*We have applied Reviewer 2's suggested method for testing the model (stated later by Reviewer*
*2), which replaces the previously used method, and are grateful for the improvement.*

*Specifics include:*

*Lines 139-147:* *"All validation data fell within the upper and lower 95% confidence intervals. It is*
*noteworthy that, for plastics longer than 100 mm, the animal-plastic size relationship under-*
*predicts the length of the longest ingested plastic, making estimates conservative. Data*
*deficiencies at both extremes of the equation means that predictions for both small (<1 mm) and*
*large (>150 mm) animals are less robust than for those between these extremes. The predicted*
*and observed data were similar (Root Mean Square of Errors (RMSE) = 0.68) and significantly*

related to one another, with reasonable explanatory power ($R^2 = 0.38$, $F_{1,5998} = 59.96$, $p < 0.001$),
suggesting predictions based on the animal-plastic size relationship are reliable.”

Lines 135-147: “Predictive power of the allometric relationship”

Lines 296-306: “Validation of the allometric relationship”

My final gripe with the study is that the authors use a case study of green sea turtles to illustrate
the utility of their “universal” equation, but don’t actually apply the equation in determining the
potential risk for turtles. I would have expected them to use the 95% CI or 99% CI from the models
to estimate the range of MP sizes that could be consumed by green turtles and then use this to
inform the cut-off for plastic concentrations in the sea water occupied by the turtles. Instead, they
use an entirely subjective size range informed by a neuston net, which has nothing to do with their
model. In other words, Figure 3 could just as easily have been produced without ever needing to
determine a relationship between animal size and minimum/maximum MPs ingested.

As was the case for Reviewer 1, the turtle case-study seemed to cause confusion. What was
done was not communicated well enough, based on Reviewer 2’s interpretation, which is
somewhat off the mark. We would add our own criticism - the turtle example was not a great
illustration of the implications and utility of the animal-plastic relationship.

With some effort, the manuscript now presents an analysis of the global zooplankton community’s
risk of ingesting plastics. Risk maps are presented side by side that illustrate the increased
accuracy of using the animal-plastic size relationship over current physical co-occurrence
approaches. We are proud of this improvement, which is directly and immediately relevant to
global plastic pollution alleviation efforts. Zooplankton can be considered as the foundation of
oceanic food-webs, and an analysis of this scale takes the field towards quantifying the flux of
plastics into global food-webs.

This considerable amendment occurs as Box 1:

Box 1 | Application of the allometric relationship

The taxonomic generality of the animal-plastic size relationship affords an array of applications. We illustrate just one, by mapping the risk of plastics entering the base of global food webs: the zooplankton community (Fig. 4). We used the animal-plastic size relationship to select an appropriate size class of plastics the global zooplankton community may ingest; then created a risk map by combining *ingestible* plastic densities as provided by Eriksen and colleagues⁸ with zooplankton densities provided by Strömberg *et al.*²⁰. The increased accuracy of assessing the fraction of plastics zooplankton *can ingest* (Fig. 4a) can be seen clearly in comparison to the same risk map for *all* plastics in the oceans (Fig. 4b). Plastics entering the global zooplankton community have substantial potential for further trophic proliferation to a broad suite of species, including commercially important quarry. Areas of priority for mitigating the influx of plastics into global food webs include the East and South China Seas, Bay of Bengal, Black, Mediterranean and Sargasso Seas, and European coasts of the north Atlantic Ocean.

Figure 4 | Global risk of plastic entering basal zooplankton communities. (a) Using the animal-plastic size relationship to estimate the density of *ingestible plastics* (0.33-1.00 mm in length) divided by zooplankton density. (b) Current approach: as for a, with *ingestible plastic* densities substituted with *total plastic* densities. Legend coarsely estimates the level of plastic ingestion risk for zooplankton.

I would recommend that the authors at the very least combine their two datasets (i.e. 66 data
points) to describe more of the variability in the true relationship between animal size and MPs
ingested. I suspect they will need to do a lot more work to build on this database over time,
however. A consideration that could help them push this into a higher tier would be to start
generating their own data by testing the range of MP particle sizes that various animals could
ingest. This would remove the troublesome doubt about whether these studies actually describe
the true range of particle sizes that each animal species can ingest.

*As mentioned previously - we are grateful to Reviewer 2 for this suggestion, which we have*
*implemented. By making better use of predictive techniques we were able to combine datasets.*
*Regarding the suggestion to generate data from artificial dosing, we feel one of the main strengths*
*of this study is that it utilises data from the natural environment. We purposefully discounted*
*laboratory-based studies, because they usually treat animals with unrepresentative*
*concentrations and size distributions of plastics. We believe studies of wild animals feeding in the*
*natural environment provide the most relevant data.*

See

*Lines 225-229: "To be considered for the *Data collection* phase (below), an article was required*
*to meet the following criteria: (1) Article seemed like it included some information on ingestion of*
*any type or size of plastic by an organism; (2) article must report on field-based studies where*
*plastics were present in the environment at natural concentrations and size distributions, as lab*
*studies are often non-representative in terms of plastic availability."*

Some minor comments:

Ln65. Change to “its body length”

**Editorial.**

Ln89. The reference to Figure 2b does not seem appropriate here. This is a figure of animal length
to smallest plastic particle ingested, not a figure summarising the use of microscopes during gut-
content surveys as the text suggests.

**Reference to Figure 2b removed.**

Ln102. This sentence entirely repeats the one at Ln89 and again, the citation to Figure 2b is
completely inappropriate.

**Amended.**

Ln108. If there is a significant relationship, then you should visualise the line of best fit (with
confidence intervals) in Figure 2b, even if the model explains a small amount of variability.

**We have implemented this suggestion:**

**Figure 3 | Detection limits scale with animal size.** Studies of larger animals tended not to
 specify the use of magnifying equipment. Weak relationship (log10-log10; $R^2 = 0.10$, $F_{1,61} = 7.58$,
 $p = 0.008$), between animals and the smallest piece of ingested plastic found during gut surveys
 (plastics measured along their longest axes).

Ln116. I assume the 95% CI here are meant to be 19–82, rather than the odd [19, 82] term that
 looks more like a citation? Why not use these 95% CI (or even 99% CI) to determine the risk of
 green turtles ingesting plastic, rather than the completely arbitrary dimensions of a neuston net
 (Ln283)? Or more correctly, you should be using the upper 95% CI of the maximum MP size

model and the lower 95% CI of the minimum MP size model to determine the plausible range of
MP particles that could be consumed by the turtles.

**As mentioned, an entirely new case-study is presented which resolves this issue.**

Ln166. Change to “short-term”

**Editorial.**

Table 1. Choose whether to present the 95% CI or 99% CI throughout and stick with that. It makes
no sense to present both. In fact, it damages your case here because we can clearly see that
most of these data either fall outside the 99% CI or barely scrape into them (and not the 95% CI).

**As mentioned, we have fully embraced Reviewer 2's suggestion for testing the model, which**
**resolves this issue.**

Reviewers' Comments:

Reviewer #1:

Remarks to the Author:

This is an interesting manuscript that is much improved for the extra additions and depth of analysis. The main figure is now quite compelling. This is an extremely important area and the results suggesting and testing an allometric relationship add to the broad discussion of how to predict bioaccumulation into the food web. I still don't quite understand the depth part, but assume that this is illustrating that surface models can't predict bioaccumulation potential whereas ingestion by animals that are capable of inhabiting greater depths could? There should be a bit more discussion of this to clarify, since how would one tell whether the plastic was ingested at depth or from the surface. Some further words of clarification would help. There is still quite a bit of repetition in the discussion and deleting this would give room to add some further information.

Reviewer #2:

Remarks to the Author:

I have read the response to reviewers' comments on the original manuscript by Jâms et al. and their associated revisions. First, let me say that this is a comprehensive consideration of the reviewers' comments and the authors have quite eloquently argued their major points of rebuttal. I think there are some excellent improvements in the revised manuscript, particularly the production of a risk map for ingestion of plastics by zooplankton (Box 1 and Figure 4), which replaces the confusing turtle example. This nicely illustrates the utility of their linear model for predicting the maximum size of plastic particles that animals can ingest in the wild.

I retain much of my scepticism about the quality of the relationship that is presented in Figure 1. Perhaps I am a "glass half empty" kind of person, but an r-squared value of 0.42 is not very convincing for me. The fact that the model dramatically underestimates the maximum size of particle ingested by large animals (the likely targets of conservation efforts) is particularly worrying to me (and again, note the log scale in the figure). Having said that, I take the point that any predictive model (even one that performs weakly) is an improvement on the current qualitative situation and that the field of plastics research needs to progress from a descriptive science to a predictive one.

I also appreciate the author's efforts to bolster their dataset and to raise confidence that the maximum particle detected in the diets relates to the maximum particle size that can be ingested (by scaling the size of the data points to the number of individuals assessed in each case). Most of the data points in Figure 1 seem quite small though...can they indicate what the minimum number of individuals examined was for including a data point? I presume they had a cut-off, e.g. they did not include a data point if <20 individuals were examined. Any less than that and it would seem unlikely that you have sampled enough individuals to quantify the maximum particle size the species could ingest. In fact, I imagine the threshold for being confident about this should be much higher, e.g. 100 individuals?

Ultimately, I would reiterate my belief that this is a well-written paper with a nice underlying idea. If the data were stronger, I would be much more enthusiastic about it. I'll let the editor decide if more enthusiasm is needed to meet the expectations of a Nature Communications audience.

REVIEWERS' COMMENTS:

Reviewer #1 (Remarks to the Author):

This is an interesting manuscript that is much improved for the extra additions and depth of analysis. The main figure is now quite compelling. This is an extremely important area and the results suggesting and testing an allometric relationship add to the broad discussion of how to predict bioaccumulation into the food web. I still don't quite understand the depth part, but assume that this is illustrating that surface models can't predict bioaccumulation potential whereas ingestion by animals that are capable of inhabiting greater depths could? There should be a bit more discussion of this to clarify, since how would one tell whether the plastic was ingested at depth or from the surface. Some further words of clarification would help. There is still quite a bit of repetition in the discussion and deleting this would give room to add some further information.

We thank Reviewer 1 for the time and effort they have invested in this manuscript, and we agree that the study has greatly benefited from the improvements suggested.

Reviewer 1 presents the idea of building on the allometric relationship presented in Figure 1 to include additional life-history variables – in this case – depth of habitat. We find this encouraging and are buoyed by the prospect of developing more sophisticated models based on animal body size as a broadly available variable. We hope the publication of this work will motivate further ambitions in this direction.

For the time being, however, we hope that tracked changes (in the Discussion especially) have clarified that Figure 2 simply draws attention to the fact that the knowledge of global plastic pollution distribution is largely confined to the surface of the Earth's oceans, and that there is a need for empirical studies at greater ocean depths (as well as terrestrial environments). This facet of the study merely highlights a research need; it is not part of the allometric relationship.

Reviewer #2 (Remarks to the Author):

I have read the response to reviewers' comments on the original manuscript by Jâms et al. and their associated revisions. First, let me say that this is a comprehensive consideration of the reviewers' comments and the authors have quite eloquently argued their major points of rebuttal. I think there are some excellent improvements in the revised manuscript, particularly the production of a risk map for ingestion of plastics by zooplankton (Box 1 and Figure 4), which replaces the confusing turtle example. This nicely illustrates the utility of their linear model for predicting the maximum size of plastic particles that animals can ingest in the wild.

I retain much of my scepticism about the quality of the relationship that is presented in Figure 1. Perhaps I am a "glass half empty" kind of person, but an r-squared value of 0.42 is not very convincing for me. The fact that the model dramatically underestimates the maximum size of particle ingested by large animals (the likely targets of conservation efforts) is particularly worrying to me (and again, note the log scale in the figure). Having said that, I take the point that any predictive model (even one that performs weakly) is an improvement

on the current qualitative situation and that the field of plastics research needs to progress from a descriptive science to a predictive one.

I also appreciate the author's efforts to bolster their dataset and to raise confidence that the maximum particle detected in the diets relates to the maximum particle size that can be ingested (by scaling the size of the data points to the number of individuals assessed in each case). Most of the data points in Figure 1 seem quite small though...can they indicate what the minimum number of individuals examined was for including a data point? I presume they had a cut-off, e.g. they did not include a data point if <20 individuals were examined. Any less than that and it would seem unlikely that you have sampled enough individuals to quantify the maximum particle size the species could ingest. In fact, I imagine the threshold for being confident about this should be much higher, e.g. 100 individuals?

Ultimately, I would reiterate my belief that this is a well-written paper with a nice underlying idea. If the data were stronger, I would be much more enthusiastic about it. I'll let the editor decide if more enthusiasm is needed to meet the expectations of a Nature Communications audience.

We are also very grateful for the time Reviewer 2 has invested in this manuscript; and agree that the work has been markedly improved by implementing the suggestions made.

It is highly encouraging that Reviewer 2 shares our appetite for advancing the field of plastic pollution research into the domain of predictive ecology. As now stated more clearly in the text; we also agree caution is needed when interpreting the relationship at the extreme ends of the body-size scale and the limitations of this work for direct conservation efforts. Specifically, see "Predictive power of the allometric relationship" in the Results section for inclusion of Reviewer 2's recommendations.

When designing the study, we decided to include any applicable data points. That is, in response to Reviewer 2's point: there was no cut-off. The study makes the best use of the available data, which includes single animals. On balance, we preferred having these data informing the trend, as opposed to having valuable (and rare) data ignored. Details appear under the sub-heading "Data collection", in the Methods section (included below).

We reiterate our statement from the previous review stage: timely publication of this study is important to encourage the publication of raw data and images alongside empirical studies of plastic pollution. In some cases, we had to work quite hard to gain the information needed – for example by using image analysis software to measure illustrative photographs of ingested plastic. Often, many more animals were included in a study than were represented by the data required.

As Reviewer 2 rightly pointed out at a previous review stage – the number of published studies of plastic pollution is expected to grow rapidly in the coming years. It is important that the opportunity isn't lost to highlight the synergistic potential of separate empirical studies that make their data available to perform meta-analyses. The additional understanding this practice affords often cuts across geographies and taxonomies. We trust the high visibility of Nature Communications will be of notable benefit in this endeavour.

We thank Reviewer 2 for their kind words and very useful feedback. We look forward to advancing the discussion around this pertinent issue.

**

Reasoning employed at the study design phase for not including a 'cut-off', taken from Methods under "Data collection":

"Articles that provided plastic and animal size data for each specimen within a study were relatively sparse. More common were summary statistics for a group of individuals of the same species. Therefore, we prioritised the collection of data on animal species (i.e. groups of individuals). Where data were available for individual animals within a group as well as for the group as a whole, only data for the latter were retained to avoid pseudoreplication. Any data on individual animals were summarised for the lowest ranked taxonomic group possible. Data on single or smaller groups of individuals of greater taxonomic resolution was prioritised over summarised data for higher taxonomic levels.

Where data were available only for subgroups where different plastic measurements were made (e.g. in surface area in some individuals or in lengths for others), we used only data expressed as linear length. Where data were available only for a single animal, summary statistics for a group were replaced with the actual values recorded for that individual. The number of animals in a group was recorded. Where data on an animal were repeated in more than one study, we used the most precise data available only to avoid pseudoreplication."

**

** See Nature Research's author and referees' website at www.nature.com/authors for information about policies, services and author benefits

This email has been sent through the Springer Nature Tracking System NY-610A-NPG&MTS

Confidentiality Statement:

This e-mail is confidential and subject to copyright. Any unauthorised use or disclosure of its contents is prohibited. If you have received this email in error please notify our Manuscript Tracking System Helpdesk team at <http://platformsupport.nature.com>.

Details of the confidentiality and pre-publicity policy may be found here <http://www.nature.com/authors/policies/confidentiality.html>

Privacy Policy | Update Profile

DISCLAIMER: This e-mail is confidential and should not be used by anyone who is not the original intended recipient. If you have received this e-mail in error please inform the sender and delete it from your mailbox or any other storage mechanism. Springer Nature America, Inc. does not accept liability for any statements made which are clearly the sender's own and

not expressly made on behalf of Springer Nature America, Inc. or one of their agents. Please note that neither Springer Nature America, Inc. or any of its agents accept any responsibility for viruses that may be contained in this e-mail or its attachments and it is your responsibility to scan the e-mail and attachments (if any).